# Copper Amine Oxidase (CuAO)-Mediated Polyamine Catabolism Plays Potential Roles in Sweet Cherry (*Prunus avium* L.) Fruit Development and Ripening

**DOI:** 10.3390/ijms232012112

**Published:** 2022-10-11

**Authors:** Xuejiao Cao, Zhuang Wen, Chunqiong Shang, Xiaowei Cai, Qiandong Hou, Guang Qiao

**Affiliations:** 1Key Laboratory of Plant Resource Conservation and Germplasm Innovation in Mountainous Region (Ministry of Education), College of Life Sciences/Institute of Agro-Bioengineering, Guizhou University, Guiyang 550025, China; 2Institute for Forest Resources & Environment of Guizhou/College of Forestry, Guizhou University, Guiyang 550025, China

**Keywords:** sweet cherry, CuAO, polyamine catabolism, *PavCuAO* gene, fruit ripening

## Abstract

Copper amine oxidases (CuAOs) play important roles in PA catabolism, plant growth and development, and abiotic stress response. In order to better understand how PA affects cherry fruit, four potential *PavCuAO* genes (*PavCuAO1*–*PavCuAO4*) that are dispersed over two chromosomes were identified in the sweet cherry genome. Based on phylogenetic analysis, they were classified into three subclasses. RNA-seq analysis showed that the *PavCuAO* genes were tissue-specific and mostly highly expressed in flowers and young leaves. Many *cis*-elements associated with phytohormones and stress responses were predicted in the 2 kb upstream region of the promoter. The *PavCuAOs* transcript levels were increased in response to abscisic acid (ABA) and gibberellin 3 (GA_3_) treatments, as well as abiotic stresses (NaCl, PEG, and cold). Quantitative fluorescence analysis and high-performance liquid chromatography confirmed that the Put content fell, and the *PavCuAO4* mRNA level rose as the sweet cherry fruit ripened. After genetically transforming *Arabidopsis* with *PavCuAO4*, the Put content in transgenic plants decreased significantly, and the expression of the ABA synthesis gene *NCED* was also significantly increased. At the same time, excessive H_2_O_2_ was produced in *PavCuAO4* transiently expressed tobacco leaves. The above results strongly proved that *PavCuAO4* can decompose Put and may promote fruit ripening by increasing the content of ABA and H_2_O_2_ while suppressing total free PA levels in the fruit.

## 1. Introduction

Polyamine (PA) is a class of plant growth regulators that are involved in the entire process of plant development, from germination to senescence. The common types of PA are putrescine (Put), spermidine (Spd), and spermine (Spm). The catabolic pathways of PA are mainly mediated by diamine oxidase (DAO) and polyamine oxidase (PAO). In plants, the CuAO can be divided into two groups depending on subcellular localization: the first group is situated in peroxisomes, based on peroxisomal targeting signals, and the other group is located extracellularly, based on N-terminal signal peptides [1]. It has been shown that members of CuAO located to the peroxisome and extracellular compartments are found in several species, including *Arabidopsis* [2], apple [3], and sweet orange [1]. In most cases, Put can be catabolized by CuAO [4], but Put and Spd can be catalyzed by CuAO in *Arabidopsis* [2]. However, PAO are involved in the terminal catabolism of Spd and Spm. At the initiation of fruit growth, PA content is at maximum, and it declines as the fruit matures and senescence occurs. Adding exogenous PA or increasing endogenous PA through genetic engineering can effectively delay fruit ripening and senescence [5,6]. Recent reports indicate that the silencing of *PpCuAO4* in peach can inhibit Put degradation, slow down the rate of fruit softening, and reduce ethylene release, suggesting that *PpCuAO4*-mediated Put decomposition can promote peach fruit ripening [7]. Studies have shown that PA can interact with hormones such as indole acetic acid (IAA) and ABA to regulate fruit ripening [8]. Additionally, complexes of metabolism involving ABA, ethylene, glycine butyric acid (GABA), etc. are associated with the PA pathway [9].

Fruits can be categorized as climacteric or non-climacteric types, based on whether respiration products and ethylene production rise throughout ripening [10]. The climacteric fruit is mainly regulated by ethylene for fruit ripening, whereas non-climacteric fruit ripening is mainly regulated by ABA [11,12]. Cherry is typical of non-climacteric fruit, and the hormone that plays a dominant role in fruit ripening in non-climacteric fruit is ABA. Long-term studies have shown that ABA can induce *CuAO* to participate in ABA signaling [13], where Spm and Spd, especially Spm, play an important role in fruit ripening, and both *FaSAMDC* and Spm/Spd positively affect fruit ripening through an ABA-dominated approach, suggesting that ABA and PA play an important role in fruit ripening [8]. An important factor, ERF, plays a significant role in controlling the ripening of fruit and is controlled by hormonal pathways like ethylene [14]. In peach, the ethylene-responsive factor *PpeERF2* regulates fruit ripening by regulating ABA biosynthesis and cell wall degradation [15]. Ethylene and ABA promote fruit ripening in grapes, while growth hormones antagonize hormones such as ABA and ethylene, delaying processes associated with ripening [16]. In grape, blueberry, and strawberry, exogenous ABA spraying increased endogenous ABA and soluble sugar content and decreased titratable acid content in the fruit, while a large number of anthocyanins accumulated in the peel, accelerating fruit coloration [17,18,19]. Meanwhile, exogenous ABA treatment promotes the expression of *NCED*, a key enzyme in the ABA synthesis pathway, which affects the ripening process of fruits [20]. The ABA content tends to increase during the ripening development of fruits such as peach and bananas [17,21]. 

The role of PA in relation to fruit has been reported several times. In olives, reduced Spd and Spm synthesis caused fruit abscission [22], while the expression levels of genes related to peach ripening were significantly reduced after exogenous Spd treatment [23]. It has also been suggested that PAO and CuAO produce H_2_O_2_ as a signaling substance to advance the fruit ripening process [24], or that PA metabolism promotes ripening by affecting the regulation of phytohormones such as ABA [7]. Moreover, grapefruit enlargement and aroma production are impacted by PA inhibitor application, which reduces peach fruit ethylene release and softening, delaying fruit ripening [14,25]. In strawberries, when *FaPAO5* is inhibited, it increases the content of ABA, Spm, and Spd, reduces H_2_O_2_ production, and effectively promotes fruit ripening [26]. PA catabolism is involved in hormonal and biological stress responses [9]. In peach, *PpPAO4* can be significantly induced by ABA treatment, and in addition, most *PAO* are induced by cold stress. Previous studies have shown that PA may play a catabolic role during peach fruit ripening by affecting genes related to hormone synthesis, signal transduction, and cell wall rupture [14]. These studies suggest that PA catabolism plays an important role in fruit ripening.

Sweet cherry (*Prunus avium*) is perishable after maturity and difficult to store, which seriously affects its economic value. Fruit ripening is a process of physiological and biochemical changes in fruit color, flavor, texture, aroma, and nutritional properties [27]. Sweet cherry is a non-climacteric fruit, and the regulatory mechanism of PA metabolism during fruit ripening has not been reported yet. In this study, the *PavCuAO* genes family was identified based on the whole genome of sweet cherry. Subsequently, the expression patterns of the *PavCuAOs* under different tissues and hormonal (ABA, GA_3_) and abiotic stress (NaCl, PEG, Cold) treatments were analyzed. During fruit ripening, PA content and the expression abundance of the *PavCuAO4* gene showed opposite trends. The function of the *PavCuAO4* gene to metabolize Put was also verified by transgenic and transient expression. In the *PavCuAO4* transgenic line, the expression of the ABA synthesis gene *AtNCED* was significantly elevated, revealing that *PavCuAO4*-mediated PA metabolism might enhance ABA biosynthesis. Our study provided further insight into the role of *PavCuAOs* in the catabolism of PAs and fruit ripening.

## 2. Results

### 2.1. Identification and Gene Structure Analysis of PavCuAOs

A total of four possible *CuAO* genes were identified in sweet cherry and named *PavCuAO1-4* (Table 1). The ORF length of the four genes ranged from 1944 bp (*PavCuAO3*) to 2181 bp (*PavCuAO4*), with isoelectric points ranging from 5.7 (*PavCuAO4*) to 7.66 (*PavCuAO3*) and molecular weights of 72.38 kDa (*PavCuAO3*) to 81.43 kDa (*PavCuAO4*). All the proteins of the four *PavCuAOs* were non-hydrophobic proteins, *PavCuAO1-3* were all localized to chromosome 1, and *PavCuAO4* was localized to chromosome 5.

The corresponding gene structure pattern was obtained by comparing the DNA sequence of the *PavCuAO* gene with the CDS sequence (Figure 1). *PavCuAO1* and *PavCuAO2* had the same gene structure, with 5 exons and 4 introns, respectively, while *PavCuAO3* and *PavCuAO4* genes both had 4 exons and 3 introns. Unlike other genes, the *PavCuAO3* gene only had a 3′ untranslated region and no 5′ untranslated region structure. 

### 2.2. PavCuAOs Promoter Analysis

Using the online website Plantcare to analyze the promoter of the *PavCuAO* gene and the *cis*-acting elements of its upstream 2000 bp sequence (Figure 2), we identified many response elements in the *PavCuAOs* promoter that are associated with phytohormones and stresses of adversity. The analysis revealed that all *PavCuAO* genes contained light-responsive element, anaerobic induction element, and auxin-responsive element, with the light-responsive element being the most prevalent. The gibberellin-responsive, ABA-responsive, and defense- and stress-responsive elements were identified in the regions of the *PavCuAO1*, *PavCuAO2*, and *PavCuAO3* promoters. The promoters of *PavCuAO1* and *PavCuAO2* also had MeJA-responsive elements, and the *PavCuAO2* and *PavCuAO3* promoters contained salicylic acid-responsive elements. In addition, *PavCuAO4* may be involved in low-temperature and wound stress.

### 2.3. PavCuAOs Protein Multiplex Sequence Alignment and Phylogenetic Analysis

The results of the protein sequence alignment for the *PavCuAOs* revealed that the protein sequences of the group’s members were highly conserved (Figure 3), with *PavCuAO1* and *PavCuAO2* sharing the maximum similarity (69.79%) and *PavCuAO2* and *PavCuAO4* having the lowest similarity (46.88%). According to the phylogenetic analysis, the *PavCuAO* genes were found to be divided into three classes. Among them, *PavCuAO1* and *PavCuAO2* were clustered into class III, and *PavCuAO3* and *PavCuAO4* were clustered into class II and class I, respectively. The result of the phylogenetic analysis showed that *PavCuAOs* and *PpeCuAOs* were closer (Figure 4).

### 2.4. Expression Analysis of PavCuAOs in Different Tissues and Fruits at Different Developmental Stages

The expression pattern of *PavCuAO* genes in different tissues of cherry (fruit, flower, stem, leaf) had obvious tissue specificity (Figure 5). Among them, the expression of *PavCuAO1* was the highest at the full flowering stage; *PavCuAO2*, *PavCuAO3*, and *PavCuAO4* were expressed to the maximum in the pre-flowering buds, young leaves, and color-breaking fruits, respectively.

The expression level of *PavCuAO1* was low, and the change was not obvious in the fruit development stage. However, *PavCuAO3* showed a slow downward trend from G1 to G4. The expression changes of *PavCuAO2* and *PavCuAO4* were identical, with the expression levels first rising and subsequently falling during the fruit ripening. The *PavCuAO2* expression level gradually decreased after reaching the peak in G2, and the expression of *PavCuAO4* first increased to the peak and then decreased in G3 (Figure 5). 

### 2.5. Expression Patterns of PavCuAO Genes under ABA, GA_3,_ and Abiotic Stress (NaCl, PEG, and Cold) Treatments

The expression levels of *PavCuAOs* under phytohormone treatment and abiotic stress were analyzed. After the ABA treatment (Figure 6), *PavCuAO1-4* all showed a clear tendency to rise and decrease with time, and *PavCuAO1*, *PavCuAO3,* and *PavCuAO4* all reached a maximum at 3 h and then decreased. In contrast, *PavCuAO2* showed a tendency to rise again after decreasing. After the GA_3_ treatment, *PavCuAO1*, *PavCuAO2,* and *PavCuAO3* all displayed the same trend of first increasing, then reducing, and finally increasing again (Figure 6). Among them, *PavCuAO1* and *PavCuAO3* both reached the maximum at 2 h, and *PavCuAO2* and *PavCuAO4* peaked at 3 h. Unlike the remaining three genes, *PavCuAO4* did not tend to rise again at 5 h.

After the NaCl stress treatment (Figure 7), the mRNA abundance of *PavCuAO1* was only slightly changed, except for the induction at 4 h. Both *PavCuAO1* and *PavCuAO2* were up-regulated in expression, while there were no significant changes at 0 h and 8 h. In contrast, *PavCuAO3* was not significantly induced throughout the stress. In contrast, *PavCuAO4* showed significant down-regulation during this process and reached its lowest value at 8 h.

For the simulated drought conditions (Figure 7), the transcript levels of *PavCuAO1* increased significantly at 4 h, with no significant changes at the remaining time points. *PavCuAO2* showed a down-regulated trend, but the expression was up-regulated at 8 h. In contrast, both *PavCuAO3* and *PavCuAO4* showed a significant up-regulated expression trend; however, *PavCuAO4* had a reduced transcript level compared to the beginning.

The expression levels of *PavCuAO1-4* genes all changed significantly under cold treatment conditions (Figure 7), accompanied by different degrees of up-regulated expression. *PavCuAO1*, *PavCuAO2*, and *PavCuAO3* all showed an overall increasing trend during cold treatment, while *PavCuAO4* showed an overall decreasing trend until 3 d, and then showed a significantly up-regulated expression at 3 d.

### 2.6. Detection of PA Content in the Ripening Process of Sweet Cherry Fruit

The content of PAs during fruit development was measured, and the results showed Put, Spd, and Spm were higher in the early stage of fruit development and decreased during the whole process. Spd and Spm decreased significantly from G2 to G3 and reached their lowest values in G4. However, the content of Put showed a stable trend after a sharp decrease from G2 to G3, while there was no significant difference between G3 and G4 (Figure 8).

### 2.7. Genetic Transformation of Arabidopsis thaliana with PavCuAO4 and Detection of Free PAs in Transgenic Plants

To understand the role of *PavCuAOs* in the catabolism of PAs, *PavCuAO4*, which is highly expressed during fruit maturation and development, was selected for the genetic transformation of *Arabidopsis thaliana*. A total of six transgenic lines were obtained, and the transcription levels of the *PavCuAO4* gene in the transgenic line were subsequently analyzed. The results showed that the *PavCuAO4* expression abundance was the highest in OE2 and the lowest in OE5 (Figure 9).

Moreover, the results of the determination of PA content in transgenic lines showed that Put content in OE5 plants was significantly lower than WT, while Spd and Spm contents were higher than those of WT. The content of PA in OE2 plants was not significantly different from those in the WT. (Figure 10).

### 2.8. Expression Patterns of Genes Related to PA and ABA Synthesis in Overexpressed Plants

The transcript levels of genes related to PA and ABA synthesis showed that the expression of PA synthesis-related genes was not significantly different between the OE2 and WT lines. In comparison to WT, OE5 dramatically increased the transcript levels of *AtADC1*, *AtADC2*, and *AtSPDS2*, and *AtSPDS1* expression levels were also enhanced. In addition, the expression level of the ABA synthesis gene *AtNCED* in OE5 was also significantly higher than that in WT (Figure 11).

### 2.9. H_2_O_2_ In Situ Histochemical Staining

The H_2_O_2_ content was analyzed by DAB staining. The results showed that the leaves on the injected side of the *PavCuAO4* gene were darker, indicating that *PavCuAO4* accumulated more H_2_O_2_ while decomposing polyamines (Figure 12).

## 3. Discussion

The catabolism of PA is mainly catalyzed by PAO and CuAO genes. At present, there are few reports on the regulation of fruit ripening by CuAO-mediated PA metabolism in sweet cherry. Currently, the most research on CuAOs is being conducted in *Arabidopsis* [28,29], followed by studies on grape [25], apple [3], tomato [30], sweet orange [1], strawberry [26], and peach [31], all of which have also been reported successively. During the rapid growth and ripening stages of tomato fruit, transcription levels related to PA biosynthesis and catabolism genes were highly expressed [30]. In peach, *PpCuAO4*-mediated Put catabolism can induce fruit ripening [31]. After peaches were treated with PA catabolism inhibitors, the fruit appeared to take longer to mature and soften, and less ABA, ethylene, and cell wall-related gene expression were seen [14].

In this study, four *PavCuAO* genes were identified through the sweet cherry genome, and phylogenetic analysis revealed that *PavCuAOs* and *PpeCuAOs* were the closest relatives. Analysis of the transcript levels of *PavCuAOs* in different tissues and at different stages of fruit development showed that *PavCuAOs* genes were significantly tissue-specific, a result that was consistent with those in peach and sweet orange [1,32], implying that *PavCuAOs* may function in different tissues. *PavCuAO1* and *PavCuAO3* were highly expressed in flowers and young leaves, whereas *PavCuAO2* and *PavCuAO4* had high transcription levels in fruits, flowers, and young leaves, suggesting that *PavCuAOs* may regulate the development of multiple organs. Both *PavCuAO4* and *PavCuAO2* displayed the same trend of change during fruit development, indicating the possibility that they may, to some extent, play the same regulatory role in fruit growth, although more research is required to confirm findings. Furthermore, PA catabolism can be induced by hormonal and biological stress responses [9], and in *Arabidopsis*, Put oxidation by *AtCuAOγ1* induced a ROS-dependent SA signaling pathway, which in turn, initiated defense systems in plants [33]. In response to ABA and GA_3_, we identified that the genes for *PavCuAO1-4* were significantly upregulated, indicating that *PavCuAOs* were implicated in these responses. In addition, most *PavCuAO* genes were induced by NaCl, PEG, and cold treatments, indicating their ability to function in different stress conditions.

Numerous studies have shown that as the fruit ripens, the PA concentration falls [30,34]. In our study, the Put, Spd, and Spm content of sweet cherry fruit decreased significantly in the G2 and G3 stages. Several studies have indicated that CuAO decomposes Put and cadaverine (Cad), albeit less effectively than Spm and Spm [35]. The results of the transgenic plants in this study showed that the Put content of OE5 was significantly lower than that of WT, so *PavCuAO4* had the function of catabolism of Put. Ethylene induces climacteric fruit ripening, while the regulation of non-climacteric fruit ripening is mainly dominated by ABA [18,20,36]. PA metabolism affects the regulation of various phytohormones such as ABA and ethylene [7], while in most cases, Put and ABA have a positive feedback mechanism that promotes each other’s synthesis [37,38]. The expression of *AtNCED* in overexpressing plants OE5 was significantly higher than that in WT. The results showed that *PavCuAO4* could promote ABA synthesis, and *PavCuAO4* synthesis was enhanced in the stage of fruit development, thereby promoting ABA synthesis. As per previous studies, CuAO-derived H_2_O_2_ may also function as a signaling chemical to speed up the ripening of fruit [38], and H_2_O_2_ is essential for ABA-mediated stomatal closure [39]. Furthermore, *CuAO* has been shown to indirectly alter NO levels by oxidizing PA to produce H_2_O_2_ to affect plant development [40], while NO and ABA can also regulate the fruit-ripening senescence process [41,42]. By using a transient expression system to inject *PavCuAO4* into tobacco leaves and measuring the H_2_O_2_ content using DAB staining, we were able to see that the gene-injected side had a deeper color than the empty side, which meant that *PavCuAO4* had acquired more H_2_O_2_ throughout the process.

During tomato development, Put content gradually rose to its peak at ripening, while Spd and Spm content declined to their respective lowest levels at this time, resulting in a reduction in the amount of total free PA in the fruit [30]. As the sweet cherry fruit ripened, the concentration of Spd and Spm decreased, which was comparable with the findings for tomato. Elevated PA content was linked to the delayed fruit ripening caused by jasmonic acid in peach fruit [43], whereas transgenic tomato leaves and ripe fruits had increased Spd content but no rise in Spm level, greatly delaying overall fruit ripening and nutritional maturity [6]. The Spm and Spd contents of transgenic plant OE5 were higher than that of WT. In particular, the Spd content was significantly higher than in WT. To explain the increased Spm and Spd contents in transgenic lines, we further analyzed the expression of PA synthesis genes in transgenic lines. The results showed that the expression of *AtADC1*, *AtADC2*, *AtSPDS1*, and *AtSPDS2* were higher in OE5 than in WT. Therefore, the genetic transformation of *PavCuAO4* enhanced PA synthesis in transgenic lines, and the Spm and Spd contents increased. While the enhanced Put synthesis was also decomposed by the overexpressed *PavCuAO4*, its content decreased. However, for some reason, the Spm and Spd content decreased during fruit ripening, while *PavCuAO4* decomposed the Put, thus reducing the total PA content in the fruit and facilitating fruit ripening. Further evidence that PA catabolism is the primary reason for the decrease in PA content during ripening originated from the upregulation of CuAO and PAO transcriptional levels during grape ripening [7]. Therefore, high levels of Spd and Spm content during fruit development may be detrimental to ripening.

Based on the reported role of PAs in fruit ripening [25,31], and by preventing fruit respiration and thus delaying the loss of soluble solids and titratable acids of fruit hardness, the PA pretreatment can enhance the storage quality of grapes [5]. Similarly, PA treatment of blueberries can also improve quality and extend storage time [44]. The *PavCuAO4*-mediated PA catabolism can promote fruit ripening, and it can also increase fruit PA accumulation and delay ripening senescence by external application of PAs. This study provides a new theoretical basis for the relationship between PAs and fruit ripening, and also provides new theoretical guidance for fruit breeding and regulation of fruit marketing.

## 4. Materials and Methods

### 4.1. Materials 

The experimental materials were collected from a 5-year-old sweet cherry plantation in the Baiyi District, Guizhou Province. The plantation was managed under regular field management. The different tissue samples included fruit (G1 for young fruit, G2 for medium fruit, G3 for color-breaking fruit, and G4 for ripe fruit), flower buds (H1: dormant bud; H2: pre-flowering bud; H3: bellflower; H4: blooming flower), leaves (Y1: young leaf; Y2: mature leaf), and stems (J1). The samples were stored in liquid nitrogen and brought back to store at −80 °C. Five-year-old sweet cherry trees in good growth condition and with healthy branches from the same parts were collected. For NaCl treatment, branches were immersed in 100 mM NaCl solution and leaves were collected at 0, 2, 4, 6, and 8 h after treatment. For cold treatment, branches were inserted into culture bottles containing water and placed in an artificial climate chamber, with a photoperiod set to 14 h of light/10 h of darkness and a temperature set to 4 °C. Samples were taken at 0, 1, 3, 6, 12, 24, 48, and 72 h after treatment. For drought treatment, branches were immersed in 20% PEG 6000 and sampled at 0, 2, 4, 6, and 8 h after treatment. For hormone treatment, cherry leaves were sprayed with 100 mg/L of ABA and GA_3_, at a dosage of 10 mL per branch, and sampled after 0, 1, 2, 3, 4, and 5 h. All treatments were collected from the leaves and stored at −80 °C. The above experiments were set up with three biological replicates.

### 4.2. Determination of PA in Sweet Cherry Fruit and Transgenic Plants

Cherry fruits and leaves of transgenic and wild-type plants were collected to detect PA content. The steps were as follows: grind the material to a powder using liquid nitrogen and weigh 0.5 g of the sample in a 10 mL centrifuge tube; add 1.5 mL of pre-cooled perchloric acid, shake well and mix, then put it into the refrigerator at 4 °C for 1 h; then centrifuge at 4 °C, 15,000 rpm for 30 min; take 1 mL of supernatant and add 10 μL of benzoyl chloride and 1 mL of 2 mol/L sodium hydroxide and mix well, then put it into the water bath at 37 °C for 30 min; add 2 mL of saturated sodium chloride and 2 mL of ether and mix well and centrifuge at 6000 rpm for 5 min; take 2 mL of ether phase, blow dry with a nitrogen blowing instrument; add 2 mL of ether, shake well, and centrifuge for 20 s, then blow dry with nitrogen; add 1000 μL of 60% methanol and shake centrifuge for 30 s; and filter with a 0.45 μm organic filter membrane and fix the volume in a 1.5 mL brown volumetric flask for testing. The determination of polyamines was performed by a high-performance liquid chromatograph (Agilent 1260, Agilent, American) on a chromatographic column (Nova-Pak C18, WATERS, American) with the mobile phase of methanol: water (60:40) at a flow rate of 1 mL·min^−1^, column temperature of 30 °C, a wavelength of 254 nm, and experiments were set up with three biological replicates. 

### 4.3. Identification of PavCuAO Gene Family Members

Based on the CuAO HMM model (PF01179) in the Pfam database, the *CuAO* family members of sweet cherry were identified as candidate genes from the whole sweet cherry genome using HMMER3. Then, the Conserved Domain Database of NCBI was used to identify conserved structural domains, and genes containing the structural domain of CuAO were retained. Referring to the method in the reference [45], the evolutionary tree construction and kinship analysis were performed using the maximum likelihood method in MEGA, multiplex protein sequence alignment was performed using DNAman, and gene structure and chromosomal localization were analyzed using TBtools [46]. The *cis*-acting elements of the promoter were predicted using the PlantCARE online website (http://bioinformatics.psb.ugent.be/webtools/plantcare/html/ (accessed on 10 August 2022)) and mapped using TBtools. Isoelectric points and relative molecular masses were calculated using ExPasy (https://web.exasy.org/protparam/ (accessed on 6 August 2022)).

### 4.4. Gene Expression Analysis

Total RNA was extracted from different tissues, treated samples, and transgenic plants, according to the RNA extraction kit method (SENO, Zhangjiakou, China), and the concentration and purity of the extracted RNA were measured by a spectrophotometer. The cDNA was then synthesized using the RevertAid First Strand cDNA Synthesis Kit (Thermo, Waltham, MA, USA) (see instructions for details) and stored at −80 °C for backup. The PA and ABA synthesis genes (*AtADC1*, *AtADC2*, *AtSPDS1*, *AtSPDS2*, *AtSPMS*, and *AtNCED*) from KEGG and the *Arabidopsis* Tair database were selected for expression analysis of transgenic *Arabidopsis thaliana*. *PavCuAO1-4* primers were designed by Primer Premier 5, and the *PavEF* gene was selected as the internal reference gene [47]. The relative expression was calculated by the 2^−ΔCt^ method [48]. The qPCR system was a 10 μL system, including 5 μL SYBR^®^ Premix Ex Taq™ II (Bemix, Dongguan, China), 2 μL diluted cDNA, 2 μL ddH_2_O, and 0.5 μL of each primer. They were performed on the qTOWER3G (Analytikjena, Jena, Germany). The reaction procedure was 50 °C for 5 min, 94 °C for 3 min, followed by 94 °C for 20 s and 60 °C for 30 s. Thirty-nine cycles were set up, with three technical replicates for each sample. The gene name and primers were listed in Appendix A. 

### 4.5. Genetic Transformation and Transient Expression

The plant expression vector was selected as pCambia1301-35s-*GFP*, as shown in Appendix A, using the Nco I and Xba I double digestion vector. According to the kit (Ready-to-Use Seamless Cloning Kit, NO. B632219, Sangon Biotech, Shanghai, China) protocol, the ORF full length of *PavCuAO3* was cloned, and the recombinant expression vector was constructed using the seamless cloning method. 

Thereafter, *Arabidopsis thaliana* was transformed using the inflorescence infestation approach utilizing the recombinant vector pCambia1301-35s-*PavCuAO4,* and the transformed seeds were harvested once they had reached maturity. The seeds were spread on MS plates containing thaumatin to screen transgenic plants and identified using PCR. Finally, the transgenic plants were transplanted into soil, and T3 generation transgenic plants were used for subsequent experimental analysis. The primers for gene clone and transgenic line identification are listed in Appendix A. In addition, the bacterial solution was injected into tobacco leaves for transient expression, the trans-null was injected as control, and the treated tobacco H_2_O_2_ staining reference [49], with slight modifications, was performed as follows: the injected tobacco leaves were immersed in the configured diaminobenzidine (DAB) solution, sealed and placed in an 80 rpm shaker overnight, and then the DAB staining solution was replaced with a bleach solution (ethanol:acetic acid:glycerol = 3:1:1) and placed in boiling water. Following the removal of the chlorophyll in a water bath and changing the bleach solution multiple times to turn the leaves white, the brown precipitate left over from the reaction of DAB and hydrogen peroxide was then photographed against a white background. 

### 4.6. Data Analysis

*T*-tests were performed with SPSS 21.0 software (Chicago, IL, USA) and plotted with Origin 9.0 (Northampton, MA, USA).

## 5. Conclusions

In summary, a total of four possible *CuAO* genes (*PavCuAO1-4*) were identified in sweet cherry. The majority of *PavCuAOs* was induced by ABA, GA_3_, and abiotic stress (NaCl, PEG, cold) treatments. The PA content of transgenic plants and transient expression analysis showed that *PavCuAO4* was involved in Put catabolism, and it may also promote fruit ripening by increasing ABA and H_2_O_2_ content and reducing the total free PA content in the fruit. This data provides a strong basis for understanding the role of the *CuAO* gene in sweet cherry development, but further experiments are needed to understand its specific role.

## Figures and Tables

**Figure 1 ijms-23-12112-f001:**
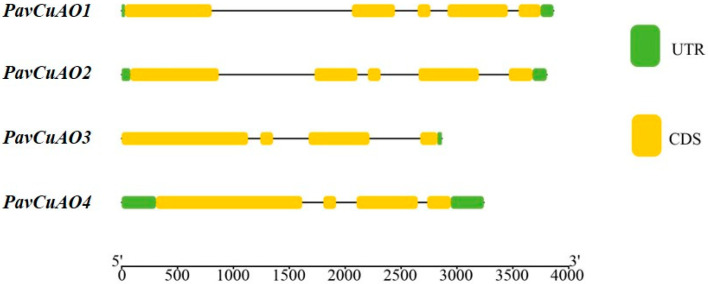
Gene structure of *PavCuAOs* in sweet cherry.

**Figure 2 ijms-23-12112-f002:**
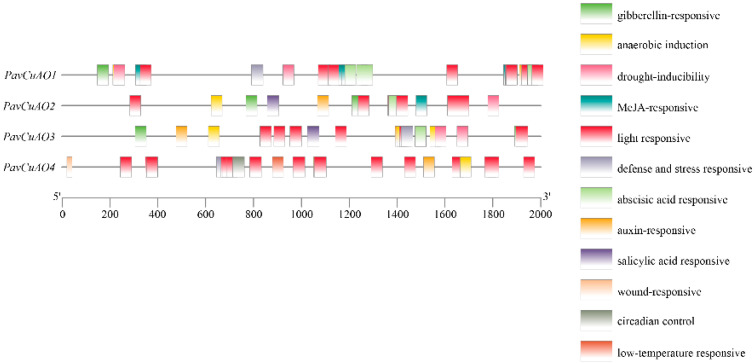
Analysis of *cis*-elements in *PavCuAOs* promoters.

**Figure 3 ijms-23-12112-f003:**
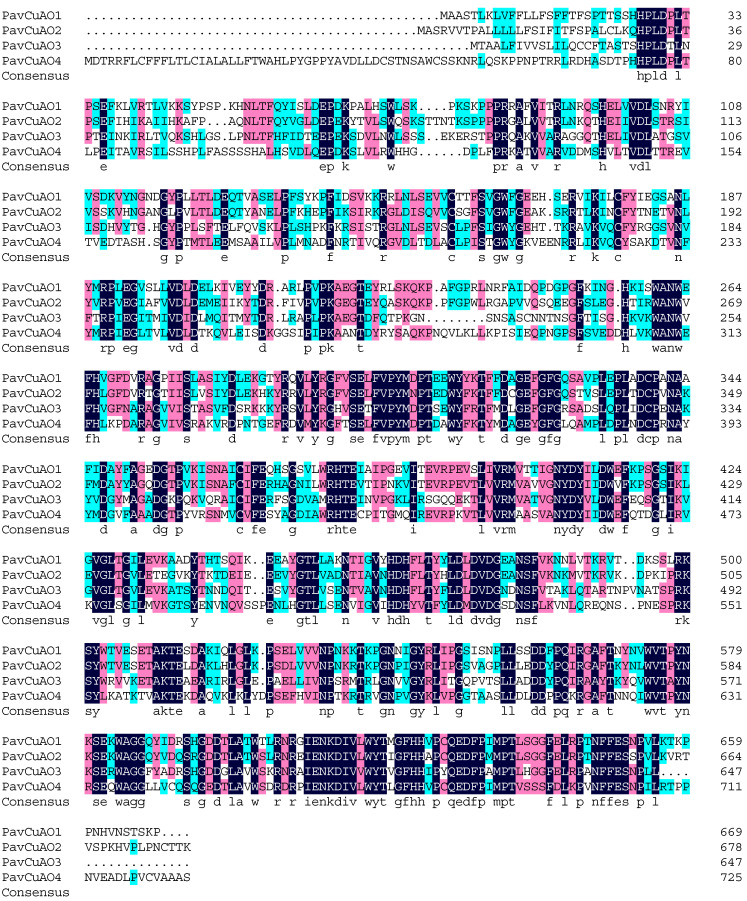
Amino acid sequence comparison of *PavCuAO* genes. (Black, red, and blue backgrounds indicate 100%, 75%, and 50% amino acid sequence similarity, respectively).

**Figure 4 ijms-23-12112-f004:**
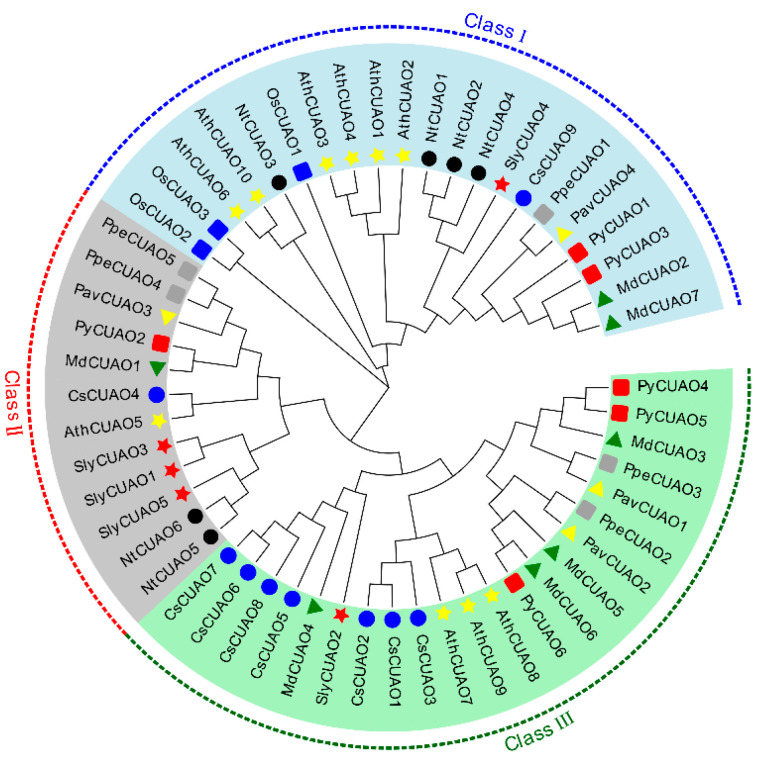
Phylogenetic tree of CuAO proteins from sweet cherry (Pav), peach (Ppe), *Arabidopsis* (Ath), tobacco (Nt), apple (Md), tomato (Sly), pear (Py), citrus (Cs), and rice (Os).

**Figure 5 ijms-23-12112-f005:**
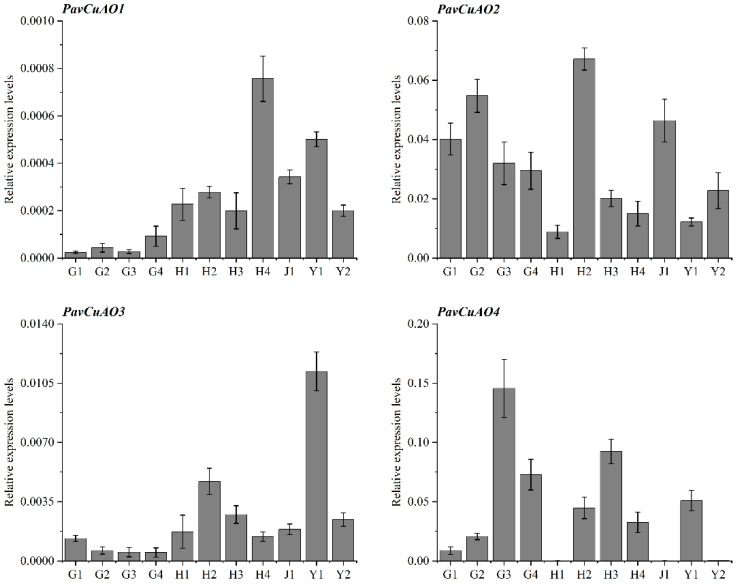
Expression patterns of *PavCuAOs* in different tissues and in the developmental stages of sweet cherry. (Note|G1: young fruit; G2: medium fruit; G3: color-breaking fruit; G4: ripe fruit; H1: dormant bud; H2: pre-flowering bud; H3: bellflower; H4: blooming flower; J1: stem; Y1: young leaf; Y2: mature leaf).

**Figure 6 ijms-23-12112-f006:**
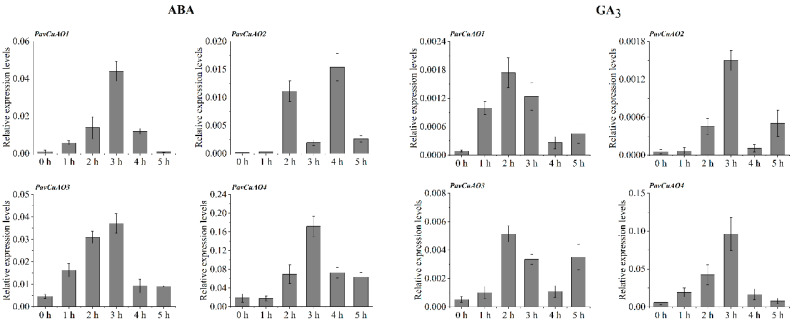
Expression pattern of *PavCuAOs* under ABA and GA_3_ treatment.

**Figure 7 ijms-23-12112-f007:**
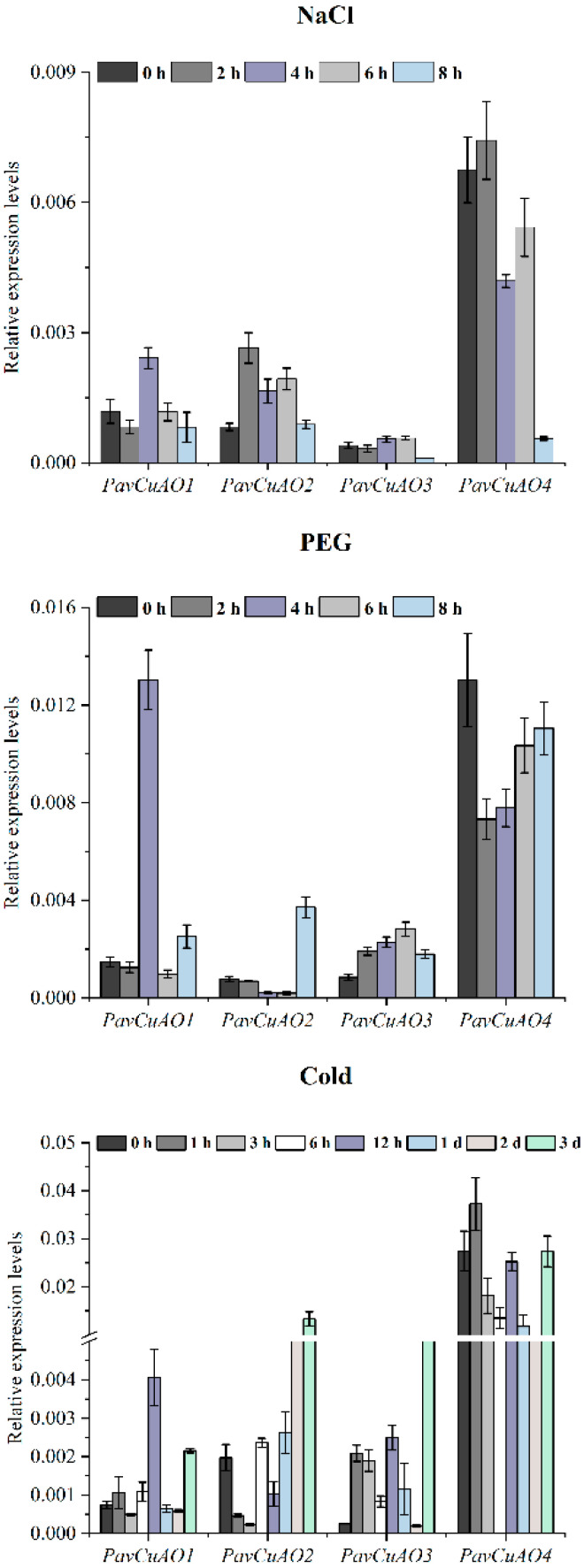
Expression patterns of *PavCuAOs* under NaCl, PEG, and cold treatment.

**Figure 8 ijms-23-12112-f008:**
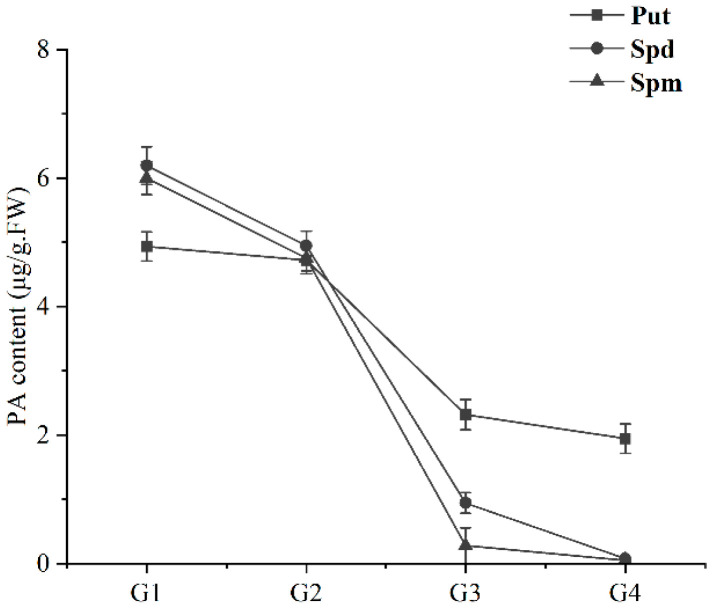
The PA content during fruit development of sweet cherry.

**Figure 9 ijms-23-12112-f009:**
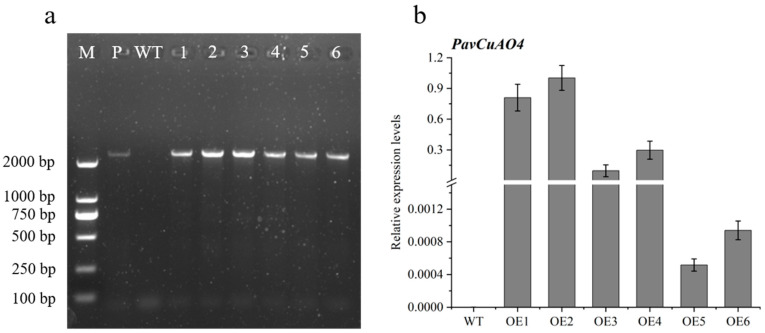
Transgenic line identification and expression analysis: (**a**) Identification of *PavCuAO4* transgenic plants (Note|M: marker; P: plasmid; WT: wild type; 1-6: transgenic plants); (**b**) transcription level of *PavCuAO4* in transgenic *Arabidopsis*.

**Figure 10 ijms-23-12112-f010:**
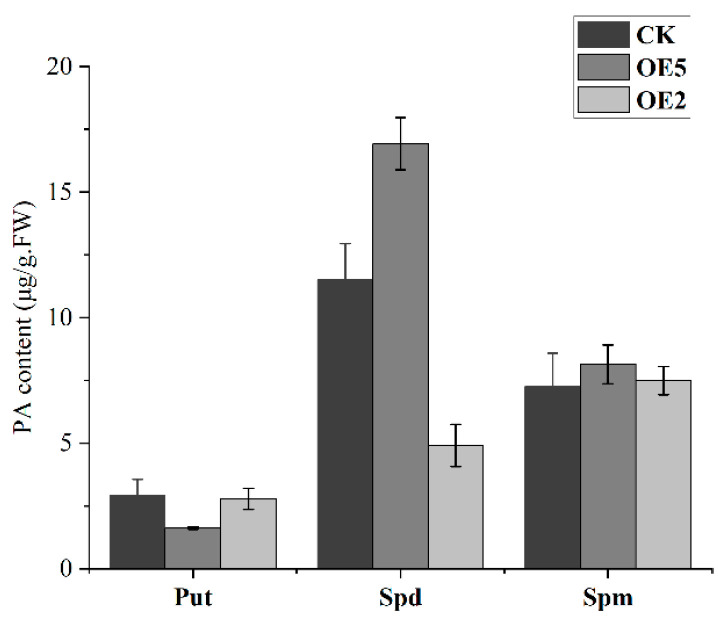
PA content of *Arabidopsis* leaves overexpressing *PavCuAO4*.

**Figure 11 ijms-23-12112-f011:**
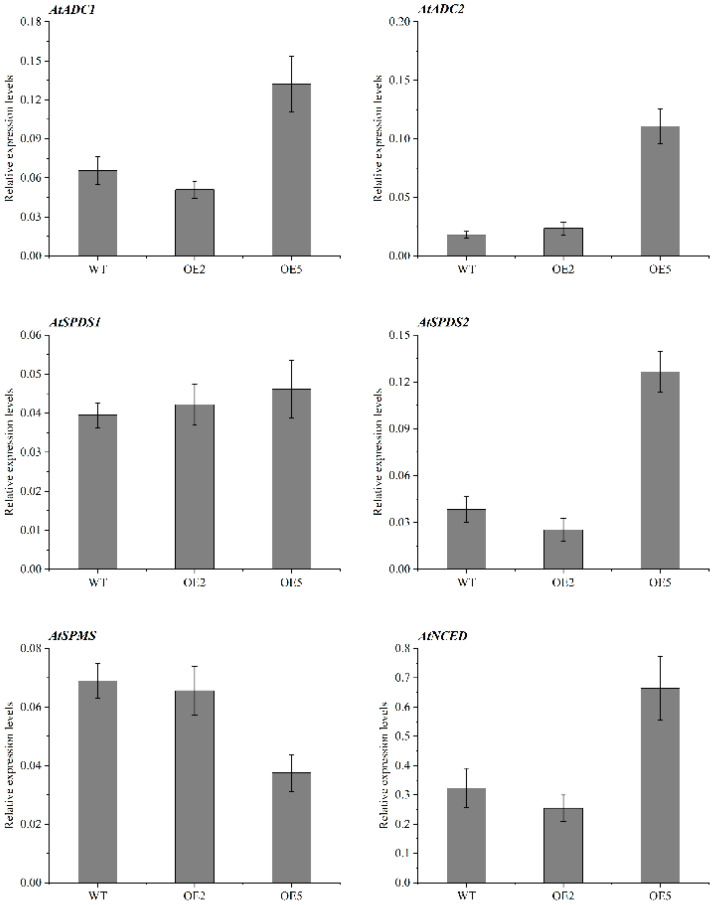
Transcription levels of PA and ABA synthesis genes in overexpressed plants.

**Figure 12 ijms-23-12112-f012:**
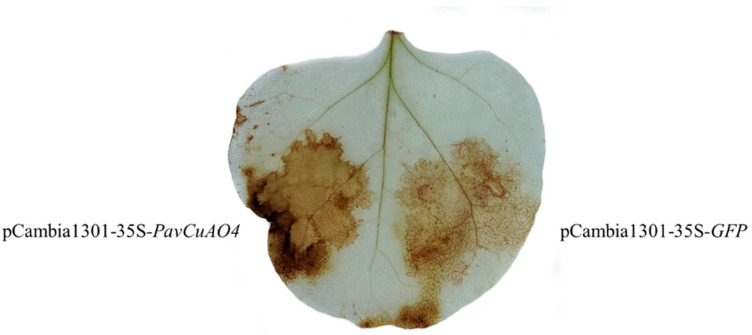
H_2_O_2_ content of tobacco leaves with transient expression of *PavCuAO4*.

**Table 1 ijms-23-12112-t001:** Description of the *PavCuAO* genes.

Gene Name	Gene ID	GRAVY	CDS Length (bp)	PI	MW (kDa)	Chromosome LOCATION
*PavCuAO1*	FUN_000165-T1	−0.306	2010	6.41	75.71	1
*PavCuAO2*	FUN_000166-T1	−0.239	2040	6.82	76.86	1
*PavCuAO3*	FUN_000167-T1	−0.276	1944	7.66	72.38	1
*PavCuAO4*	FUN_024869-T1	−0.268	2181	5.7	81.43	5

## Data Availability

Not applicable.

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
