# Peer review of "Copper Amine Oxidase (CuAO)-Mediated Polyamine Catabolism Plays Potential Roles in Sweet Cherry (Prunus avium L.) Fruit Development and Ripening"

_ijms, 2022, doi:10.3390/ijms232012112_

Round 1

Reviewer 1 Report

Dear Authors, 

You should improve the introduction by giving a solid research problem statement by mentioning information from all the relevant articles, especially those that work based on genome-wide studies. 

The write-up needs improvement and please check carefully and avoid such sentence ''DAO is a copper-containing enzyme that catabolizes Put [5],'' at the start of the sentence to prevent the use of abbreviation. 

The methodology should be elaborated on in a little bit detail. 

Dear Authors, 

You should improve the introduction by giving a solid research problem statement by mentioning information from all the relevant articles, especially those that work based on genome-wide studies. 

The write-up needs improvement and please check carefully and avoid such sentence ''DAO is a copper-containing enzyme that catabolizes Put [5],'' at the start of the sentence to prevent the use of abbreviation. 

The methodology should be elaborated on in a little bit detail. 

Please mention the criteria for the selection of genes for qPCR.

Please use the Maximum likelihood tree instead of NJ. Please see the article below:

Magnesium transporter Gene Family: Genome-Wide Identification and Characterization in Theobroma cacaoCorchorus capsularis, and Gossypium hirsutum of Family Malvaceae

The BAHD Gene Family in Cacao (Theobroma cacao, Malvaceae): Genome-Wide Identification and Expression Analysis

Please develop the discussion well by mentioning how the current study improves our knowledge and its importance. 

Author Response

Response to Reviewer 1 Comments

1) Comment: You should improve the introduction by giving a solid research problem statement by mentioning information from all the relevant articles, especially those that work based on genome-wide studies. 

Response: Thanks for your reminding and advice. We have supplemented CuAO family research and functions in lines 52-66 of the revised version.

2) Comment: The write-up needs improvement and please check carefully and avoid such sentence ''DAO is a copper-containing enzyme that catabolizes Put [5],'' at the start of the sentence to prevent the use of abbreviation. 

Response: Thanks for your reminding and advice. We have amended the similar issues in the article.

3) Comment: The methodology should be elaborated on in a little bit detail. 

Response: Thanks for your reminding and advice. We have described the article's method in detail.

4) Comment: Please mention the criteria for the selection of genes for qPCR.

Response: Thanks for your reminding and advice. We have added to the gene criteria used for qPCR in lines 491-493 of the revised version.

5) Comment: Please use the Maximum likelihood tree instead of NJ. Please see the article below:

Response: Thanks for your reminding and advice. We have reconstructed the evolutionary tree using the maximum likelihood method in line 190.

6) Comment: Please develop the discussion well by mentioning how the current study improves our knowledge and its importance. 

Response: Thanks for your reminding and advice. We have already discussed the significance and usefulness of this study in the discussion.

Reviewer 2 Report

Comments and Suggestions for Authors

The following are some general points and scientific queries that need to be addressed in the

Title: The title is good and appropriate for the proposed scientific research

Abstract:  We note in the Abstract that the Methodology and  Conclusion  are not mentioned in detail, which loses the originality and methodology of the research, which is the most important corner of the study. Therefore, the researcher must mention the methodology and conclusion.

Suggest the keywords: sweet cherry; CuAO; polyamine catabolism;PavCuAO genes; fruit ripening

Introduction: When I read the introduction, I found the researcher had mastered a simple and an inapprehensible way of writing the introduction so that the reader could understand an overview of CuAO–mediated polyamine catabolism plays potential roles in sweet cherry and pointed out the problem of the research as well as its goal.

Results: Describe the major findings of your study in the opening sentence. Writing results need improvement.

Discussion: Please discuss and support your study by using new references.

Materials and methods: Before starting the design of any scientific research, the scientific researcher must choose the scientific method that fits the subject of his research study.

Therefore, we advise the researcher to follow the steps of writing the scientific research in detail and clearer descriptions.

References should follow the International Journal of Molecular Sciences format.

English Proficiency: We invite him to delve deeper into the English language and master its rules.

Author Response

Response to Reviewer 2 Comments

1) Comment: The title is good and appropriate for the proposed scientific research

Response: Thanks for your suggestion.

2) Comment: We note in the Abstract that the Methodology and Conclusion are not mentioned in detail, which loses the originality and methodology of the research, which is the most important corner of the study. Therefore, the researcher must mention the methodology and conclusion.

Response: Thanks for your suggestion. We have added further explanation of the methodology and conclusions in the abstract.

3) Comment: Suggest the keywords: sweet cherry; CuAO; polyamine catabolism; PavCuAO genes; fruit ripening

Response: Thanks for your suggestion. We have made changes to the keywords.

4) Comment: Introduction: When I read the introduction, I found the researcher had mastered a simple and an inapprehensible way of writing the introduction so that the reader could understand an overview of CuAO–mediated polyamine catabolism plays potential roles in sweet cherry and pointed out the problem of the research as well as its goal.

Response: Thanks for your suggestion. We have made further adjustments to the introduction.

5) Comment: Results: Describe the major findings of your study in the opening sentence. Writing results need improvement.

Response: Thanks for your suggestion. We have revised the results for the full text. 

6) Comment: Discussion: Please discuss and support your study by using new references.

Response: Thanks for your suggestion. We have added the new references to the discussion in lines 422-430 and other place of the revised version.

7) Comment: Materials and methods: Before starting the design of any scientific research, the scientific researcher must choose the scientific method that fits the subject of his research study. Therefore, we advise the researcher to follow the steps of writing the scientific research in detail and clearer descriptions.

Response: Thanks for your suggestion. We have described and adapted the method in detail.

8) Comment: References should follow the International Journal of Molecular Sciences format.

Response: Thanks for your suggestion. We have checked the format of the references.

9) Comment: English Proficiency: We invite him to delve deeper into the English language and master its rules.

Response: Thanks for your suggestion. We have had the article revised by native English speakers.

Round 2

Reviewer 1 Report

No comment